# Genome-Wide Identification of the *Oxidative Stress 3* (*OXS3*) Gene Family and Analysis of Its Expression Pattern During Ovule Development and Under Abiotic Stress in Cotton

**DOI:** 10.3390/biology13110903

**Published:** 2024-11-06

**Authors:** Yu Chen, Rui Yang, Haojie Wang, Xianghui Xiao, Baoguang Xing, Yanfang Li, Qiankun Liu, Quanwei Lu, Renhai Peng, Guodong Chen, Yongbo Wang, Pengtao Li

**Affiliations:** 1School of Biotechnology and Food Engineering, Anyang Institute of Technology, Anyang 455000, China; cyu990324@163.com (Y.C.); 15388465669@163.com (R.Y.); 17516150781@163.com (H.W.); 18317714953@163.com (X.X.); 19937820150@163.com (B.X.); sybks10lyf25@163.com (Y.L.); liuthundering@163.com (Q.L.); daweianyang@163.com (Q.L.); aydxprh@163.com (R.P.); 2Xinjiang Production and Construction Corps Seventh Division Agricultural Research Institute, Kuitun 833200, China; 3College of Agriculture, Tarim University, Alar 843300, China; 4Cotton Sciences Research Institute of Hunan, Changde 415101, China

**Keywords:** cotton, OXS3, genome-wide identification, phylogenetic analysis, abiotic stress response

## Abstract

Our study comprehensively performed genome-wide analyses of the oxidative stress 3 gene family in four representative *Gossypium* species, including identifications of family members, evolutionary relationships, structural features, chromosomal locations, collinearity, cis-acting elements, and expression patterns. Furthermore, the potential functions of cotton OXS3 genes against multiple abiotic stresses were verified. These findings offer valuable insights for use in molecular breeding aimed at cotton germplasm innovation or genetic improvement.

## 1. Introduction

The OXS3 protein plays a crucial role in plants, particularly under stress conditions, when OXS3 family proteins are essential to oxidation reduction and the regulation of the stress responses to heavy metal exposure. OXS3 belongs to a protein family characterized by a highly conserved domain known as the acetyltransferase or thiol transferase catalytic domain (ACD). Based on the similarity of conserved sequences within this domain, the presence of OXS3 genes has been predicted in plant genomes. To date, the *OXS3* gene family has been identified in various species, including *Arabidopsis thaliana* [1,2], wheat (*Triticum aestivum*) [3], and rice (*Oryza sativa*) [4]. The mutations within the ACD of *Schizosaccharomyces pombe* were reported to reduce the *At*OXS3’s ability to enhance stress tolerance, indicating the significant role of this domain in stress resistance [5]. In *Arabidopsis*, the OXS3 protein might function as a histone modification factor that could actively respond to heavy metal and oxidative stress. Under drought conditions, the absence of *AtOXS3* led to the phenotype of early flowering, and it has been proved that OXS3 can bind to SOC1 (*Suppressor of Overexpression of Constans 1*), thereby inhibiting the expression of the *APETALA1* (*AP1*) gene. Since *AP1* is well-known gene participating in flower development, the suppression of *AP1* expression could mitigate stress-induced early flowering [6]. Overexpression of *OXS3* gene family members in rice was observed to significantly reduce cadmium levels in grains [7]. Both of the homologous genes of *AtOXS3* and *OsOXS3*, namely *OsO3L2* and *OsO3L3*, were found to co-localize with certain histones [4,5]. The cellular localization of *OsO3L2* and *OsO3L3* suggested that these nuclear proteins could interact with histone H2A in vascular cells, and their interaction with histone H2A might alter chromatin structures to regulate downstream gene expression [7]. The plant protein of heterologous AtOXS3 was reported to bind to components of the *Schizosaccharomyces pombe* HA2.Z and SWR1 complexes, promoting the replacement of H2A with H2A.Z. This histone replacement could increase the occupancy of the oxidative stress-response transcription factor Pap1 in the promoter, enhancing transcription and thereby providing greater stress tolerance to the cells [8].

Developmental analysis was conducted on ABA-hypersensitive mutant seedlings in *Arabidopsis*. The results indicate that OXS3, OXS3b, O3L3 (OXS3 LIKE 3), O3L4, and O3L6 might act as negative regulators of ABI4 expression. Under non-stress conditions, OXS3 family proteins, in conjunction with AFP1, could regulate the deposition of γ-H2A.X at the ABI4 promoter. By reducing the occupancy of histone γ-H2A.X in the promoter, they collectively suppressed ABI4 expression, thereby preventing ABA-induced growth arrest [1]. In plants, SNF1-related protein kinase 1 (SnRK1) sensed nutritional and energy statuses, then translated these signals into appropriate responses. There were two putative SnRK1 recognition motifs found in the ACD, and SnRK1 could interact with most of OXS3 family proteins [2]. Under stress conditions, the ABA-responsive protein SnRK1 could phosphorylate OXS3 family proteins, causing their translocation from the nucleus to the cytoplasm. This reduction of OXS3 proteins in the nucleus led to an increased occupancy of γ-H2A.X at the ABI4 promoter, thereby lifting the repression of ABI4 expression and activating the expression of downstream ABA-responsive genes. ABA is known to protect citrus from green mold caused by *Penicillium digitatum*. The absence of OXS3 proteins caused the activation of ABA-responsive genes in plants [9].

Cotton is a globally important natural fiber crop, serving not only as a crucial raw material for the textile industry, but also as an edible oil consumed by more than half of the world’s population [10,11]. As a field crop, cotton’s yield and fiber quality are susceptible to various other abiotic and biotic stress factors [12,13,14,15,16]. Up to data, 53 species within the *Gossypium* genus have been identified, comprising 46 diploid subspecies (A-G and K) and 7 allotetraploid subspecies (AD). As the most widely planted cotton species, *G. hirsutum* (Upland cotton) and *G. barbadense* (Sea Island) contribute more than 98% of fiber production. *G. arboreum* and *G. raimondii* have been proven by an increasing body of evidence to be the donor species for the tetraploid formation [17,18,19,20]. The four representative species have not only complete high-quality genome sequences but are also generally regarded as ideal models of ploidy and ideal for evolutionary analyses of the gene family. Therefore, these species were chosen for use in our research. We identified a total of 69 OXS3 family members from the genomes using genome-wide identification methods. The comprehensive analyses of physicochemical properties, phylogenetic relationships, chromosomal localization, gene structure, conserved motifs, conserved domains, and cis-acting elements were conducted on the cotton OXS3s, whose expression patterns relevant to fiber development and responding to abiotic stresses also implied their potential functions in the biological processes. These results provide a solid foundation for further research into cultivating high-yield, superior-quality, and multi-resistant cotton materials by overexpressing or gene-editing the candidate *OXS3* genes.

## 2. Materials and Methods

### 2.1. Identification of Cotton OXS3 Family Members

In this study, the genome sequences of *G. hirsutum* (AD1) and *G. barbadense* (AD2) from the COTTONOMICS Database (http://cotton.zju.edu.cn/download.html, accessed on 20 September 2024), and the genome sequences of *G. arboreum* (A2) and *G. raimondii* (D5) from the CottonFGD website (https://cottonfgd.net/about/download.html, accessed on 20 September 2024) were separately downloaded. A total of 8 *Arabidopsis* OXS3s protein sequences were obtained from NCBI and were utilized as reference sequences to perform BLAST searches against the genome files of the four representative cotton species using TBtools v2.056 software. Additionally, the cotton OXS3 proteins were identified using the PFAM database (http://pfam-legacy.xfam.org/, accessed on 21 September 2024) and HMMER [21] (https://www.ebi.ac.uk/Tools/hmmer/, accessed on 21 September 2024). Candidate sequences were further confirmed using NCBI CD-Search (https://www.ncbi.nlm.nih.gov/Structure/cdd/wrpsb.cgi, accessed on 21 September 2024) and SMART [22] (https://smart.embl-heidelberg.de/, accessed on 21 September 2024). The amino acid number, molecular weight (MW), isoelectric point (pI), and grand average of hydropathicity (GRAVY) of each of the cotton OXS3 proteins were predicted using the online analysis tool ExPASy (https://web.expasy.org/protparam/, accessed on 22 September 2024).

### 2.2. Analyses of Phylogeny, Gene Structure, Conserved Motifs, and Cis-Acting Elements

Multiple sequence alignments of the OXS3 protein sequences among *Arabidopsis* and the four cotton species were performed using the AligninMUSCLE method in MEGA X 10.1.18 software. A phylogenetic tree was constructed with the Neighbor-Joining method in MEGA X. The Partial Deletion and P-distance model was chosen and the bootstrap value was set to 1000. The tree was visualized using the Evolview website (https://evolgenius.info//evolview-v2/#login, accessed on 23 September 2024) [23]. The gene structure and conserved motifs of the OXS3 family members in the four cotton species were analyzed using TBtools [24]. To investigate the cis-acting elements of the cotton *OXS3* genes, the 2000 bp upstream sequences from the start codon were extracted. These sequences were then uploaded to the PlantCare website (http://bioinformatics.psb.ugent.be/webtools/plantcare/html/, accessed on 24 September 2024) to predict the cis-acting elements in the promoter regions.

### 2.3. Chromosome Localization and Gene Duplication Analysis of Cotton OXS3 Genes

The gff files of the four cotton species were input into the TBtools software to extract the chromosomal location information of the *OXS3* genes and the chromosome length data. Gene duplication and syntenic regions were identified using the MCScanX tool [25].

### 2.4. Analyses of Tissue-Specific, Development-Related, and Respoding-Adversity Expression

The online database COTTONOMICS (http://cotton.zju.edu.cn/, accessed on 24 September 2024) was chosen for use in this study when we performed the expression-pattern analyses of cotton’s *OXS3* genes, which was composed of the assembled genome data and multiple transcriptome data [26]. The expressing data of root, stem, leaf, petal, pistil, sepal, and torus were utilized to conduct a tissue-specific analysis on the 22 *GhOXS3* genes and the developing ovules (0, 1, 3, 5, 10. and 20 DPA). The fibers (10, 20, and 25 DPA) were also selected to investigate the development-related expression of *GhOXS3* genes. Their PFKM values were subjected to normalization processing using a Z-score algorithm, and a heatmap presenting the tissue-specific and development-related expression was drawn by TBtools with the same scaleplate.

Meanwhile, the response expression patterns of 22 *GhOXS3* genes to multiple adversities were analyzed based on the RNA-seq data of cold (4 °C), heat (37 °C), salt (NaCl), and drought (PEG) stresses. A similar statistical analysis was employed on their FPKM values, resulting in 4 separate heatmaps with the same scaleplate by TBtools.

### 2.5. RNA Extraction and qRT-PCR Analysis of Cotton OXS3 Genes

Plump TM-1 seeds were selected and planted in nutrient pots filled with a mixture of sand and vermiculite, which were placed in a greenhouse maintained at 28 °C with a photoperiod of 16 h of light and 8 h of darkness. When the cotton seedlings developed to the stage of two fully expanded true leaves, the plants were separately treated under 4 °C (cold stress), 37 °C (heat stress), 200 mM NaCl (salt stress), and 10% PEG6000 (drought stress). Subsequently, cotton leaves were collected at 0, 6, 12, and 24 h post-treatment, respectively, which were subjected to RNA extraction using a FastPure Universal Plant Total RNA Isolation Kit from Vazyme (Vazyme Biotech Co., Ltd., Nanjing, China). RNA concentration and integrity were measured with a NanoDrop 2000 (Nanjing Junwei Biotechnology Co., Ltd., Nanjing, China), and the high-quality RNAs were utilized to perform reverse transcription with a HiScript III RT SuperMix for qPCR (+gDNA wiper) kit from Vazyme (Vazyme Biotech Co., Ltd., Nanjing, China). Specific primers for the *OXS3* gene family were designed using NCBI Primer-BLAST (Appendix A). We conducted qRT-PCR reactions using ChamQ Universal SYBR qPCR Master Mix from Vazyme on an ABI 7500 Fast (Applied Biosystems, Foster City, CA, USA). The house-keeping gene *GhUBQ7* (DQ116441) was utilized as the internal reference for normalizing the relative expression levels, which was calculated by the 2^−ΔΔCt^ method [27].

## 3. Results

### 3.1. Genome-Wide Identification of OXS3 Family Members in Cotton

A total of 69 OXS3 proteins were identified across the four representative cotton species, including 22 GhOXS3, 23 GbOXS3, 12 GaOXS3, and 12 GrOXS3 proteins, all of which harbored a conserved N-acetyltransferase (ACD) domain. The numbers of OXS3 proteins in the tetraploid cottons were approximately double those found in the diploid cottons, and this phenomenon was consistent with their ploidy relationships. For convenience, *OXS3* gene family members were renamed as follows: *GhOXS3-1* to *GhOXS3-22* for *G. hirsutum*, *GbOXS3-1* to *GbOXS3-23* for *G. barbadense*, *GaOXS3-1* to *GaOXS3-12* for *G. arboreum*, and *GrOXS3-1* to *GrOXS3-12* for *G. raimondii* (Appendix A). The physicochemical properties of the OXS3 family members were subsequently predicted in the four cotton species, and the results indicate that the OXS3 proteins range in length from 96 (GbOXS3-1, GaOXS3-1, and GrOXS3-4) to 311 (GbOXS3-18 and GrOXS3-11) amino acids. The minimum and maximum values of their molecular weights were separately 11.38 kDa (GbOXS3-1) and 34.49 kDa (GbOXS3-18), and the range of their isoelectric points (pI) was from 4.85 (GhOXS3-5) to 10.3 (GhOXS3-2 and GbOXS3-2). The grand average of hydropathicity (GRAVY) values ranged from −1.286 to −0.059, indicating that most of these proteins were hydrophilic.

### 3.2. Phylogenetic and Domain Analysis of the OXS3 Gene Family

To gain deeper insights into the evolutionary relationships of the *OXS3* gene family, homologous alignment was first conducted on the protein sequences among the four cotton species and other model plant species, and a phylogenetic tree was subsequently constructed (Figure 1). The results indicate that the total of 96 OXS3 proteins from the four cotton species, *Arabidopsis*, rice, and wheat were divided into three groups (Group 1–Group 3). The largest number of OXS3 proteins from the four cotton species and *Arabidopsis* fell into Group 3. There was a lack of OXS3 proteins in wheat. The fewest OXS3 proteins were found in Group 2, which did not contain any OSX3s from *G. hirsutum* or *Arabidopsis*. The ACD acetyltransferase domain was relatively conserved among *Arabidopsis* and the four cotton species (Appendix A), indicating its crucial role in identifying the *OXS3* gene family. The differences between the cotton and other species might have resulted from the locating evolutionary position, and having more members of the gene family in cotton could provide more potential functions during development and growth as well as in responses to multiple stresses.

### 3.3. Gene Structure and Protein Motif Analysis of the OXS3 Gene Family

To further investigate the structural features within the *OXS3* gene family in cotton, the evolutionary relationships of the 69 OXS proteins were explored (Figure 2A). The results showed the same group categories clustered among the cotton OXS3 members as those divided by cotton and other species OXS3s (Figure 1). The member distribution was consistent with the above-mentioned phylogenetic tree. Subsequently, the analysis of conserved motifs of the OXS3 proteins was performed among the four representative cotton species, resulting in no more than 6 kinds of conserved motifs (Figure 2B). We noticed that all of the members of the OXS3 *gene* family, except for *GhOXS3-1*, contain Motif 1 and Motif 2. In Group 3, there were at least four motifs in most of the OXS3 members except GbOXS3-13 and GrOXS3-7. Additionally, more conserved motifs were observed in Group 3 than in Group 1 and Group 2, suggesting that functional divergence of the *OXS3* genes in cotton during evolution might lead to differences in conserved motifs among family members. Furthermore, OXS3 members within the same group exhibited relatively similar conserved motifs. These results indicate that OXS3 family members located on the same evolutionary branch might have been relatively conserved throughout their evolutionary history.

Meanwhile, the exon-intron diversities of the cotton OXS3 *gene* family were analyzed, and the results indicated most of the *OXS3* genes in cotton contain 1–2 introns, the exception being *GrOXS3-7*, which has no introns (Figure 2C). Within the same group, OXS3 genes exhibit similar exon-intron structures and intron numbers, implying highly conserved gene structures.

### 3.4. Chromosomal Localization and Gene Duplication of the OXS3 Gene Family

To visually understand the chromosomal distributions of the *OXS3* family, localization analyses were performed on all of the cotton *OXS3* genes (Figure 3). The results indicated that *OXS3* genes were located on 6 and 9 different chromosomes in diploid and tetraploid cotton, respectively, with up to 4 *OXS3* genes on a single chromosome.

Gene duplication events make great contributions to plant evolution, of which whole-genome duplication, segmental duplication, and tandem duplication are regarded as the major drivers of gene family expansion. The numbers of *OXS3* genes in tetraploid cotton were approximately double those found in diploid cotton, suggesting that gene duplication events might occur during the tetraploidization process. To detect gene duplication events in the *OXS3* gene family, the intra- and inter-species collinearity of *OXS3* genes in the four cotton species were analyzed (Figure 4 and Appendix A). The analysis of intra-species collinearity revealed a total of 103 pairs of gene duplication events across the four cotton species, including 44 pairs in *G. hirsutum*, 42 pairs in *G. barbadense*, 9 pairs in *G. arboreum*, and 12 pairs in *G. raimondii*. We noticed that the duplication events were widely found in the cotton OXS3 gene family, of which the occurrence rate in allotetraploid cotton was significantly higher than that in diploid cotton. These results indicated that OXS3 genes might have undergone member amplification in polyploid cotton. Inter-species collinearity analysis was separately conducted between the diploid A genome (*G. arboreum*) and the tetraploid A subgenomes (*G. hirsutum* and *G. barbadense*), as well as between the diploid D genome (*G. raimondii*) and the tetraploid D subgenomes (*G. hirsutum* and *G. barbadense*). There were 23 and 20 pairs of gene duplication from Ga-Gh and Ga-Gb in the former comparison, while 20 and 20 pairs of gene duplication from Gr-Gh and Gr-Gb in the latter comparison. These results suggested that whole-genome duplication, segmental duplication, and tandem duplication could play significant roles in the expansion of the OXS3 gene family in cotton. Similar numbers of duplication pairs were found between the diploid A genome and the diploid A subgenomes, as well as between the diploid D genome and the tetraploid D subgenomes.

### 3.5. Cis-Acting Elements in the Promoters of OXS3 Gene Family in Cotton

Since cis-acting elements in promoters could affect the regulation of gene expression, the 2000 bp upstream promoter sequences of the *OXS3* gene family were extracted from four cotton species and then utilized to predict the activity of the cis-acting elements. We did this to gain deeper insights into the functions of the *OXS3* genes (Figure 5). The results revealed the presence of various response elements in the promoters of the OXS3 family members across the four cotton species. The cis-acting elements included light-responsive elements, hormone-responsive elements related to abscisic acid, MeJA, salicylic acid, and auxin, and defense and stress-responsive elements associated with low temperature, drought, and anaerobic conditions. This information indicates that the *OXS3* gene family in cotton might actively respond to multiple hormonal and environmental signals, thereby participating in relevant regulatory and physiological processes.

### 3.6. Tissue-Specific and Development-Related Expression of OXS3 Genes in G. hirsutum

To screen the potential *OXS3* genes that could regulate specific biological processes, we analyzed the expression levels of the *OXS3* genes in *G. hirsutum* (*GhOXS3-1* to *GhOXS3-22*) based on the published transcriptome data. The clustering heatmap indicated that the expression patterns of *GhOXS3* genes varied across different tissues (Figure 6). In Group 1, *GhOXS3-1* was highly expressed in roots, while *GhOXS3-9* showed high expression levels in pistils. The expression levels of other *GhOXS3* genes in this group were not significantly different across tissues. In Group 3, *GhOXS3-20* was found to be highly expressed in pistils, and other *GhOXS3* genes exhibited higher expression levels in pistils, petals, sepals, and receptacles than were found in roots, stems, and leaves. The different developmental stages of ovules and fibers were also utilized to comprehend the expression levels of *GhOXS3* genes, which result in the relatively highly expressed *GhOXS3-2* found in 10 DPA fibers and the highest expression levels of *GhOXS3-1* being found in 25 DPA fibers. The other *GhOXS3* genes mainly showed high expression in ovules at 0 and 1 DPA. These results indicate that *GhOXS3* genes might harbor tissue-specific functions, which could be induced to play different roles at various developmental stages, particularly during the early stages of ovule development.

### 3.7. Expression Patterns of GhOXS3 Genes in Response to Multiple Adversity Stresses

In order to further analyze the expression patterns of *OXS3* genes under stress conditions, we turned to the existing transcriptome data relevant to the cotton samples subjected to the cold, heat, salt, and drought treatments at different time points (0 h to 24 h). After re-analyzing the expression patterns and clustering analysis, we noticed that *GhOXS3* genes within the same group exhibited similar expression patterns (Figure 7). Under drought stress, most of the GhOSX3 genes showed first up-regulated and then down-regulated expression patterns. Only *GhOXS3-5*, *GhOXS3-7*, and *GhOXS3-17* continuously increased their expression levels throughout the stress periods. The expression trends of *GhOXS3* genes responding to drought stress were similar to those found in the same genes under salt stress. Most were first up-regulated and then down-regulated, the exceptions being *GhOXS3-16* and *GhOXS3-21,* both of which presented continuously down-regulated patterns. More than half of the GhOXS3 genes showed continuously up-regulated expression patterns under clod and heat stresses. The remainder exhibited the up-regulated and then down-regulated trend. We also found that only GhOXS3-2 was not expressed under multiple abiotic stresses. These findings suggest that *OXS3* genes may play significant roles in cotton’s response to abiotic stresses.

### 3.8. Quantitative Expression Patterns of GhOXS3 Genes Under Multiple Abiotic Treatments

Five genes each of the OXS3 family members in Group 1 and Group 3 of *G. hirsutum* were randomly selected for qRT-PCR analysis to examine the expression patterns of *OXS3* genes under drought (PEG), salt (NaCl), cold (4 °C), and heat (37 °C) treatments at 0, 6, 12, and 24 h. Clustering analysis indicated that the *OXS3* genes showed diverse expression changes in response to different stress treatments (Figure 8). Under the salt treatment, the expression levels of most of the 10 *GhOXS3* genes increased. *GhOXS3-1*, *GhOXS3-5*, *GhOXS3-12*, *GhOXS3-21*, and *GhOXS3-22* showed an initial decrease followed by an increase. Under the drought treatment, the expression levels of all of the 10 *GhOXS3* genes also increased, of which *GhOXS3-3*, *GhOXS3-10*, *GhOXS3-19*, *GhOXS3-21*, and *GhOXS3-22* showed significant up-regulation. Under the cold treatment, most of the 10 *GhOXS3* genes showed increased expression levels. Only *GhOXS3-1*, *GhOXS3-10*, *GhOXS3-19*, and *GhOXS3-21* exhibited an initial decrease followed by an increase. Similarly, under the heat treatment, the expression levels of all of the 10 *GhOXS3* genes increased, with all of the genes showing elevated expression levels at 6 h and *GhOXS3-19* exhibiting the most significant up-regulation after treatment. The expression patterns of the genes in Group 1 and Group 3 were similar across these four treatments. Therefore, *GhOXS3* genes might be widely involved in the response to abiotic stresses and likely play positive roles in cotton’s resistance to such stresses.

## 4. Discussion

Environmental stress leads to crop yield reduction as both biotic and abiotic stresses disrupt the normal metabolic balance of cells, causing an increase in intracellular reactive oxygen species (ROS) levels, which in turn leads to cell damage and death [28,29]. To elucidate the molecular mechanisms by which plants respond to the oxidative damage induced by heavy metal stress, extensive research has been conducted on three genes involved in oxidative and metal stress: OXS1, OXS2, and OXS3. OXS1 encodes a conserved eukaryotic protein containing an HMG box, thereby participating in a novel OXS1-PAP1 regulatory pathway [30]. In *Arabidopsis*, OXS2 is a key regulator in the stress escape process, by which plants transition from vegetative to reproductive growth under extreme environmental conditions to ensure species survival [31]. The maize OXS2 family were found to enhance plant resistance against the heavy metal cadmium by activating a putative SAM-dependent methyltransferase-like gene [32]. In *Arabidopsis*, the OXS3 protein might act as a histone modification factor, responding to heavy metal and oxidative stress [5]. O3L3 was reported to bind to the promoter of *Arabidopsis* ZAT12, while the OXS3 family member in *Brassica napus*, namely BnKCP1 (also known as BnO3L1), exhibited moderate transcriptional activation activity [5,33,34]. Under non-stress conditions, OXS3 (including OXS3, OXS3b, O3L3, O3L4, O3L6) interacted with AFP1 to inhibit the expression of the *ABI4* gene [1].

The *OXS3* gene family was also identified by recent studies of various plant species, including eight members in *A. thaliana* [1,2], nine members in wheat [3], and ten members in rice [4]. However, the OXS3 gene family in *Gossypium* species has not yet been reported upon. Cotton is an economically important crop not only for producing natural fibers for the textile industry, but also as an oilseed crop providing nutrient-rich plant oil for human consumption [35]. Additionally, the genome sequencing and assembly of multiple cotton species with different ploidy levels have been completed, providing a solid foundation for a bioinformatics analysis of the *OXS3* gene family [36]. Therefore, four representative *Gossypium* species were selected in this study for a genome-wide analysis of the *OXS3* gene family, resulting in a total of 69 OXS3 members, including 22 GhOXS3s, 23 GbOXS3s, 12 GaOXS3s, and 12 GrOXS3s. The numbers of OXS3 proteins in the tetraploid species were nearly double those found in diploid species, which was consistent with the typical pattern of polyploid formation. Together with the eight OXS3 proteins from *Arabidopsis* (AtOXS3s), nine from wheat (TaOXS3s), and ten from rice (OsOXS3s), all of the identified OXS3 proteins in cotton were classified into three groups (Group 1, Group 2, and Group 3) for phylogenetic analysis (Figure 1). In the analysis of OXS3 family members in *Arabidopsis* and rice, no clustering group study was performed for phylogenetic analysis [4,5]. In the phylogenetic analysis of wheat, the OXS3 family members from *Arabidopsis*, rice, and wheat were divided into three groups [3], with no *Arabidopsis* OXS3 members in Group 2, two AtOXS3s members in Group 1, and six AtOXS3s members in Group 3. The results of our phylogenetic analysis are consistent with this, with no *G. hirsutum* OXS3 members appearing in Group 2. Sequence alignment of the 8 *Arabidopsis* and 69 OXS3 gene family members from the four cotton species revealed that the key acetyltransferase (ACD) domain of the OXS3 family was highly conserved in both *Arabidopsis* and the four cotton species. In summary, the *OXS3* gene family appears to have maintained relative conservatism throughout evolutionary history and has expanded in higher plants.

The evolutionary relationships of all of the 69 cotton OXS3 genes showed the same clustering categories as those found in other modal plants (Figure 2), which were also subjected to gene-structure analysis. Notably, *GrOXS3-7* lacked introns entirely. Variations in exon/intron counts might result from alternative splicing events during transcription, providing insights into evolutionary mechanisms [37]. Genes with increased numbers of introns were often considered to acquire new functions throughout evolution [38]. Additionally, we examined the cis-acting elements in the promoter regions to predict their roles in gene transcriptional regulation. Our findings indicated the presence of various cis-acting elements in the cotton OXS3 gene family that respond to hormonal and environmental signals (Figure 5). Among all 69 OXS3 genes, the most abundant cis-acting elements are related to plant stress hormones such as SA, ABA, and MeJA. These hormones have been reported to play positive roles in plant defense against abiotic stresses [39,40,41,42]. For instance, *Arabidopsis* OXS3 family proteins were found to inhibit ABA signaling by interacting with AFP1, thereby regulating the expression of *ABI4* [1]. In citrus, the absence of OXS3 could activate the response of ABA-related genes [9].

Previous studies have suggested that allotetraploid cotton species (AADD) might have originated from the chromosomal doubling and hybridization of diploid cotton species AA (possibly *G. herbaceum*) and DD (*G. raimondii*) [43]. Representative cotton materials with different ploidy levels and improved genomic information provide an ideal model for gene duplication studies. In this study, intra- and inter-species collinearity analyses were separately conducted on *G. arboreum* (diploid A2), *G. raimondii* (diploid D5), *G. hirsutum* (tetraploid AD1), and *G. barbadense* (tetraploid AD2), identifying a total of 103 pairs of duplicated genes across the four cotton species (Figure 4). The highest number of duplicated gene pairs was observed in *G. hirsutum*, followed by *G. barbadense*, with *G. arboreum* having the fewest. These duplicated pairs include tandem duplications, segmental duplications, and whole-genome duplications, suggesting that these mechanisms might be the primary drivers of the expansion of the *OXS3* gene family in cotton. Referring to the studies of the whole-genome analysis of various plant gene families, segmental duplications were proved to contribute more significantly than tandem duplications. Examples include the MADS-Box in maize [44], CCD in tobacco [45], RabGAP in tomato [46], and EPF/EPFL in cotton [47]. Additionally, nearly identical gene duplication events were found between diploid genomes and tetraploid subgenomes, indicating that both the A and D genomes have contributed equally to the expansion of the OXS3 gene family in cotton throughout evolution.

Based on previous RNA-seq data from *G. hirsutum* TM-1 [48], the expression patterns of 22 *GhOXS3* genes were investigated across different tissues and developmental stages. Tissue-specific expression analysis revealed that certain *GhOXS3* genes, such as *GhOXS3-1*, were highly expressed in roots, *GhOXS3-9* and *GhOXS3-20* were highly expressed in pistils, and *GhOXS3-18* was highly expressed in receptacles and sepals (Figure 6). We also observed that *GhOXS3* genes exhibit relatively higher expression levels in pistils, petals, sepals, and receptacles compared to roots, stems, and leaves, suggesting that *OXS3* genes might make great contributions to promoting reproductive growth in cotton. For instance, in *Arabidopsis*, the interaction between OXS3 and SOC1 inhibited premature flower development and stress-induced flowering responses [6]. In developing ovules, except for *GhOXS3-1* and *GhOXS3-2*, other *GhOXS3* genes showed the highest expression levels at 0 and 1 DPA. During the fiber elongation and secondary wall thickening stages, only *GhOXS3-2* exhibited high expression at 10 DPA, while the expression levels of other *GhOXS3* genes did not show significant changes. Overall, although there was no conclusive evidence linking *OXS3* genes to early ovule development, these candidate genes might provide new insights into their potential functions in future studies.

Similarly, the response patterns of all of the *GhOXS3* genes to cold, heat, salt, and drought stress were analyzed using published transcriptome data (Figure 7). Among the *GhOXS3* genes under salt and drought stress, similar expression patterns were observed, with most of the *GhOXS3* genes initially showing a sustained up-regulation trend. In response to cold stress, half of the *GhOXS3* genes exhibited continuous up-regulation, while the remaining *GhOXS3* genes showed high expression at 6 h and 12 h, followed by a decrease at 24 h. Under heat stress, most *GhOXS3* genes exhibited continuous up-regulation, with the remaining genes showing high expression at 12 h and a decrease at 24 h. Given the close relationship between sulfate transport and a plant’s adaptation to temperature stress, the common gene *GhOXS3-2* may play a role in plants’ defense against various environmental factors. Previous studies have also indicated that wheat *OXS3* genes have a positive function in defending against heat stress [3], but also could make great contributions to plant resistance for withstanding fungus infection [9]. Certainly, our study mentioned only the abiotic-stress responses of cotton OXS3 genes, and their roles in responding to biotic stresses, such as *Verticillium* and *Fusarium* wilt or some insect pests, might require more transcriptome data and experimental results to verify in the future.

Furthermore, ten randomly selected *GhOXS3* genes were subjected to qRT-PCR experiments under different stress treatments (Figure 8). Except for *GhOXS3-21* under the drought treatment and *GhOXS3-5*, *GhOXS3-6*, and *GhOXS3-18* under the salt treatment, where expression levels did not show significant changes, all of the *GhOXS3* genes exhibited significant up-regulation after the four treatments. The relative expression patterns confirmed by qRT-PCR were generally consistent with the RNA-seq data, indicating the need for further validation of the *GhOXS3* gene functions through genetic transformation techniques. Based on the analysis results, the *GhOXS3-3* gene could be chosen to perform the overexpressing or knock-out experiments in *Arabidopsis* and cotton to validate its function when combined with the phenotypes of the transgenic plants under various abiotic stress conditions. We believe that the over-expressed or silenced materials will ultimately lead to the development of new cotton varieties with enhanced stress resistance.

## 5. Conclusions

In this study, a comprehensive genome-wide analysis of the *OXS3* gene family was conducted in four cotton species, categorizing 69 *OXS3* genes into three groups. Using bioinformatics tools, the gene structure, phylogeny, chromosomal localization, conserved domains, and cis-acting elements of the cotton *OXS3* genes were also analyzed. Combined with the tissue-specific, development-related, and adversity-response expression patterns, by re-analyzing the published RNA-seq data, the quantitative expression verification was performed on some candidate GhOXS3 genes, all of which showed consistent expression trends. Our findings revealed that *OXS3* genes could actively respond to different stress treatments and might be extensively involved in abiotic stress responses, potentially playing positive roles. This study provides a foundation for further exploration of the functions of *OXS3* genes in cotton.

## Figures and Tables

**Figure 1 biology-13-00903-f001:**
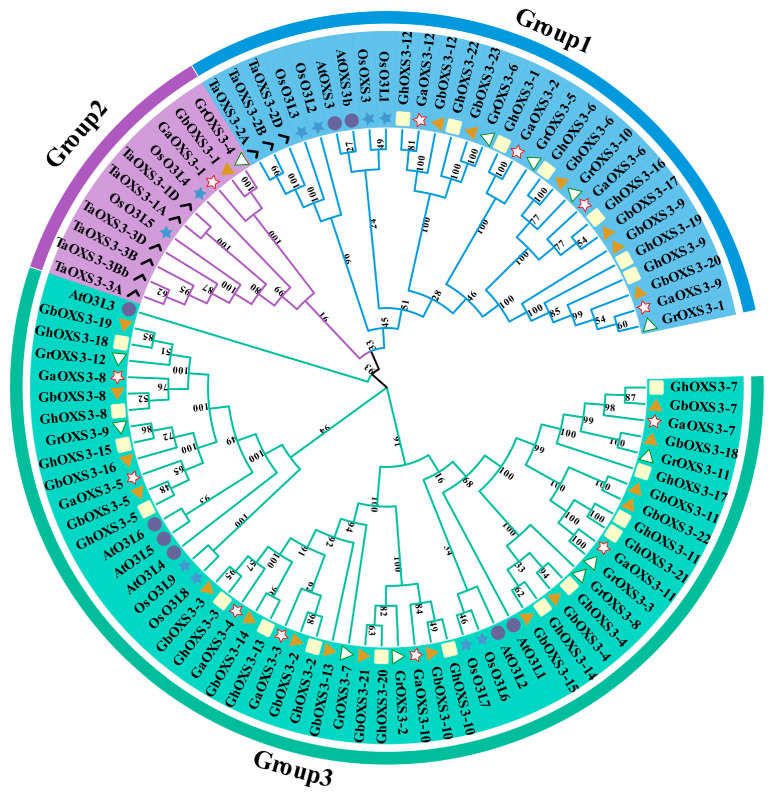
Phylogenetic tree of OXS3 proteins from the four cotton species and other model plants. Different colored arcs in the outer circle represent different groups of OXS3 proteins. Blue stars indicate OXS3 proteins from rice, black check marks represent OXS3 proteins from wheat, purple circles denote OXS3 proteins from *Arabidopsis*, yellow squares indicate OXS3 proteins from *G*. *hirsutum*, orange triangles represent OXS3 proteins from *G*. *barbadense*, red hollow stars denote OXS3 proteins from *G*. *arboreum*, and white hollow triangles represent OXS3 proteins from *G*. *raimondii*.

**Figure 2 biology-13-00903-f002:**
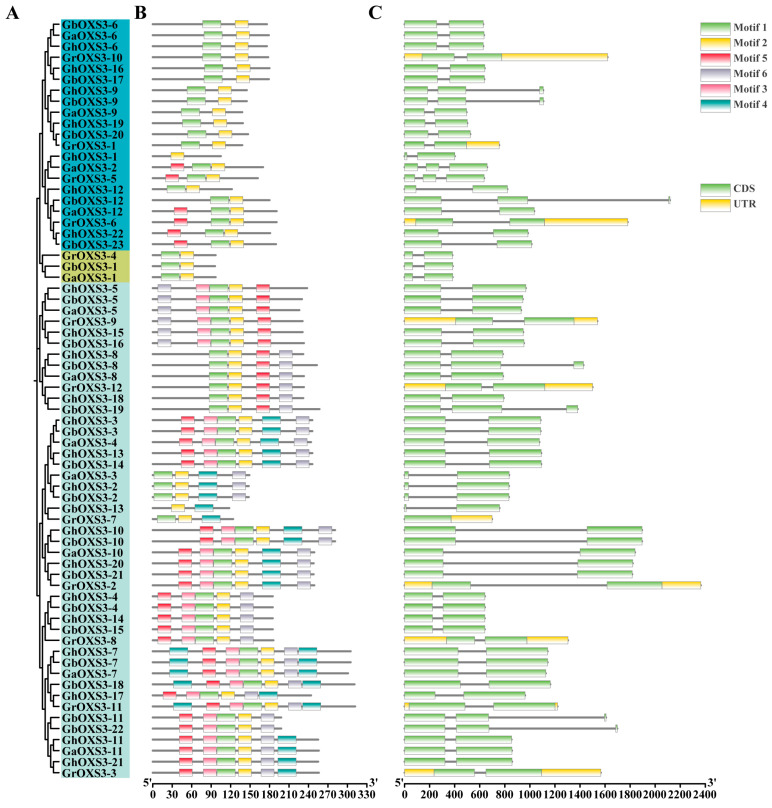
Analyses of the evolutionary relationships and structural features of cotton OXS3 proteins. (**A**) The evolutionary relationships of 69 cotton OXS3 proteins. (**B**) Conserved motifs of the *OXS3* gene family in cotton. (**C**) Gene structures of the cotton *OXS3* gene family. The lengths of protein and DNA sequences are estimated using the scale at the bottom, with black lines representing non-conserved amino acids or introns.

**Figure 3 biology-13-00903-f003:**
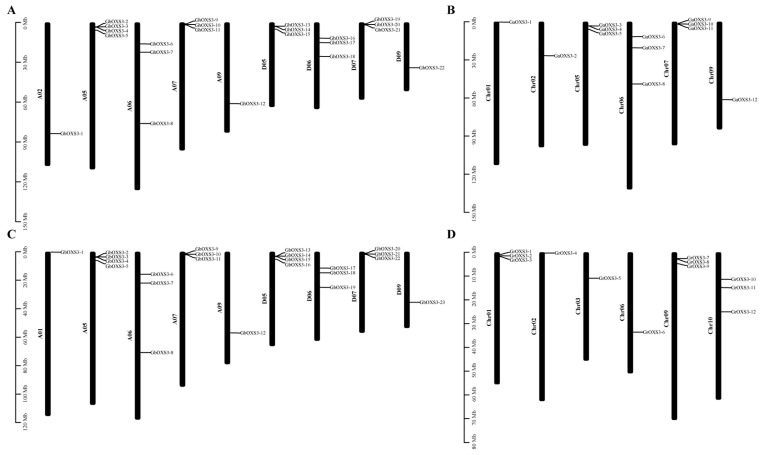
The chromosomal distribution of the *OXS3* gene family in cotton. (**A**) The chromosomal distribution of 22 *GhOXS3* genes. (**B**) The chromosomal distribution of 12 *GaOXS3* genes. (**C**) The chromosomal distribution of 23 *GbOXS3* genes. (**D**) The chromosomal distribution of 12 *GrOXS3* genes. The vertical lines on the left represent chromosome sizes, measured in megabases (Mb).

**Figure 4 biology-13-00903-f004:**
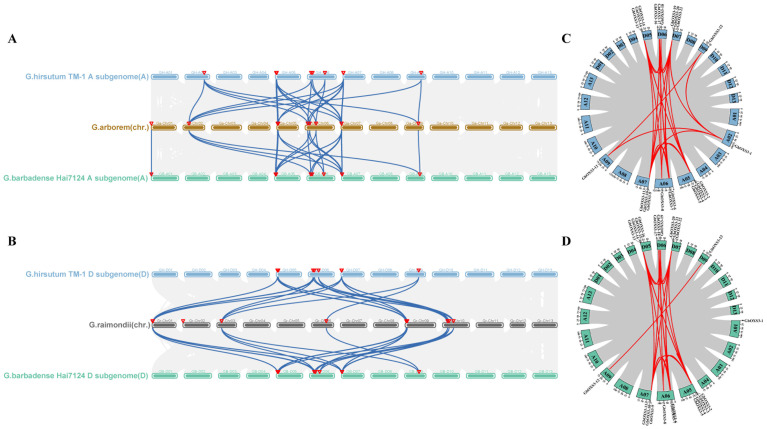
Collinearity analysis of *OXS3* genes among the genomes of four cotton species. (**A**) Collinearity between the A genome of *G. arboreum* and the A subgenomes of *G. hirsutum* and *G. barbadense*. (**B**) Collinearity between the D genome of *G. raimondii* and the D subgenomes of *G. hirsutum* and *G. barbadense*. (**C**) Collinearity analysis between the A and D subgenomes of *G. hirsutum*. (**D**) Collinearity analysis between the A and D subgenomes of *G. barbadense*.

**Figure 5 biology-13-00903-f005:**
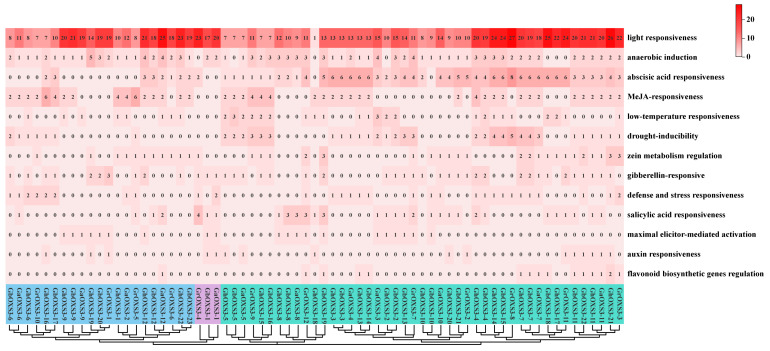
Cis-acting elements of the *OXS3* gene family in four cotton species. The numbers represent the count of cis-acting elements present in the *OXS3* gene.

**Figure 6 biology-13-00903-f006:**
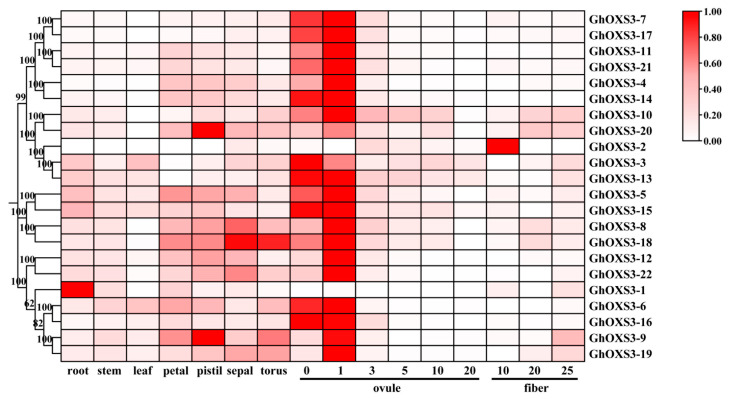
Expression levels of *GhOXS3* genes in different tissues and during various developmental stages of ovules and fibers. The darker the color, the higher the expression levels.

**Figure 7 biology-13-00903-f007:**
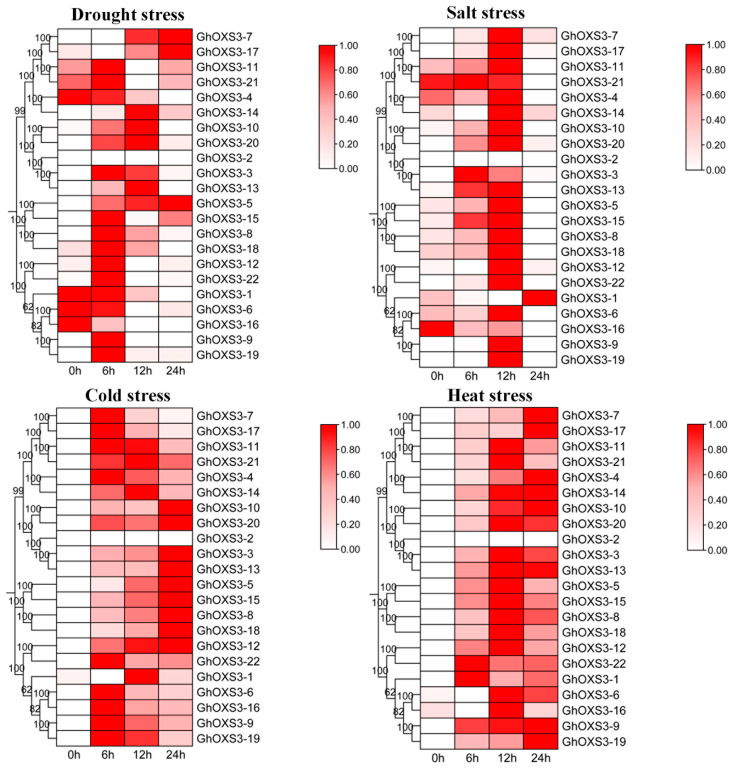
Transcriptome-expression levels of *GhOXS3* genes under adversity stresses. The darker the color, the higher the expression levels.

**Figure 8 biology-13-00903-f008:**
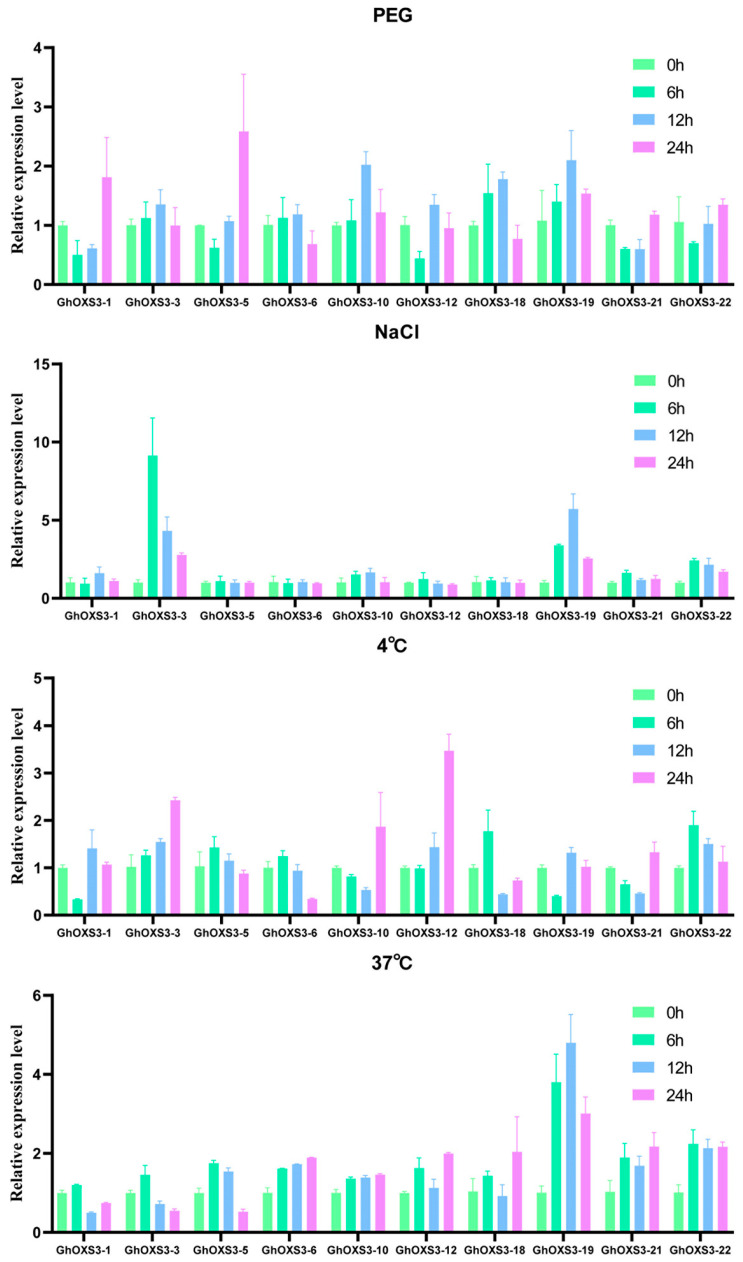
qRT-PCR analysis of *GhOXS3* expression under different stress treatments. The different colors represent the different treatment timepoints and the taller columns represent the higher expression levels.

## Data Availability

The original contributions presented in the study are included in the article/Appendix A, further inquiries can be directed to the corresponding author.

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
