# Peer review of "Genome-Wide Identification of the Oxidative Stress 3 (OXS3) Gene Family and Analysis of Its Expression Pattern During Ovule Development and Under Abiotic Stress in Cotton"

_biology, 2024, doi:10.3390/biology13110903_

Round 1
Reviewer 1 Report
Comments and Suggestions for Authors
This manuscript presents valuable information on identifying the OXS3 gene family in cotton, with implications for improving abiotic stress tolerance. The study is well-executed and supported by comprehensive data. The writing is clear and provides an insightful description of the study, including the main objectives, key findings, interpretations, and the significance of the research. However, I have some suggestions to further improve it. With these minor revisions, the manuscript would be suitable for publication.
1. The title does not accurately reflect the manuscript's content, particularly the second part, "their potential functions affecting ovule development." While you have identified the genes and examined their expression, a functional analysis of the OXS3 gene family in cotton is needed, not just an exploration of gene expression related to abiotic stress responses and ovule development. Please amend the title accordingly.
2. In the Introduction, if there are previous studies on OXS3 in other plant species beyond Arabidopsis, please include this information. It's also important to highlight the research gap and justify the need for this study.
3. Consider adding a section on statistical analysis in the Materials and Methods to show how the gene expression data were normalized and analyzed for significance.
4. The Results section is solid. However, it would benefit from a more detailed explanation of gene duplication events and their role in the expansion of the OXS3 family.
5. The phylogenetic tree (Figure 1) would be clearer with an improved color scheme to more easily distinguish between the different groups.
6. The Discussion is generally well-rounded, with appropriate interpretations of the results. However, it would broaden the study’s relevance to agricultural stress tolerance if you discuss how these findings in cotton might be applicable to other crops,
Author Response
Comments 1: The title does not accurately reflect the manuscript's content, particularly the second part, "their potential functions affecting ovule development." While you have identified the genes and examined their expression, a functional analysis of the OXS3 gene family in cotton is needed, not just an exploration of gene expression related to abiotic stress responses and ovule development. Please amend the title accordingly.
Response 2: Thank you very much for your careful and valuable reviewing. After having synthesizing all the research findings, we have revised the title of “Genome-wide identification of oxidative stress 3 (OXS3) gene family and their potential functions affecting ovule development and responding to abiotic stress in cotton” as “Genome-wide identification of oxidative stress 3 (OXS3) gene family and their expression-pattern analysis during ovule development and under abiotic stress in cotton”.
Comments 2: In the Introduction, if there are previous studies on OXS3 in other plant species beyond Arabidopsis, please include this information. It's also important to highlight the research gap and justify the need for this study.
Response 2: Thank you very much for your worthy suggestion. We have revised the sentence of “However, reports on the genome-wide identification and expression pattern analysis of OXS3 are limited” as “However, reports on the genome-wide identification and expression pattern analysis of OXS3 were only found in Arabidopsis, wheat, and rice.”
Comments 3: Consider adding a section on statistical analysis in the Materials and Methods to show how the gene expression data were normalized and analyzed for significance.
Response 3: Thank you very much for your careful and valuable reviewing, and we have added one section entitled “2.4 Analyses of tissue-specific, development-related, and respoding-adversity expression” in the revised manuscript. The details how we performed the data normalization processing and heatmap analysis on the gene expression data were described in the following sentences (Line 152-165), and we did not repeat those sentences here since they take up plenty of space.
Comments 4: The Results section is solid. However, it would benefit from a more detailed explanation of gene duplication events and their role in the expansion of the OXS3 family.
Response 4: Thank you very much for your worthy suggestion, and we have added the descriptions in the revised manuscript. A more detailed explanation of gene duplication events and their role in the expansion of the OXS3 family were stated in the line 258-262.
Comments 5: The phylogenetic tree (Figure 1) would be clearer with an improved color scheme to more easily distinguish between the different groups.
Response 5: Thank you very much for your careful and valuable reviewing. We have revised the Figure 1 to present an improved color scheme, which will be conducive more easily distinguishing between the different groups.
Comments 6: The Discussion is generally well-rounded, with appropriate interpretations of the results. However, it would broaden the study’s relevance to agricultural stress tolerance if you discuss how these findings in cotton might be applicable to other crops.
Response 6 : Thank you very much for your worthy suggestion. We have added the sentences relevant to agricultural stress tolerance in the Line 489-493, and discussed the potential candidate gene OXS3-3 for further verification in Line 498-506.
Reviewer 2 Report
Comments and Suggestions for Authors
The Authors conducted comprehensively perform of the oxidative stress gene family in four Gossypium species. A large amount of data was summarized and bioinformatics analysis of evolutionary relationships, structural features, chromosomal localizations, collinearity, cis-acting elements was performed, and expression patterns were systematically studied using databases. These studies are relevant for further practical application in cotton breeding.
A big data was analyzed, but the contribution of the authors themselves is not always clear, for example, Figure 6: the data provided by the authors is a bioinformatics analysis of databases or data from a database is presented. Or in the case of qRT-PCR analysis of GhOXS3 expression under different stress conditions. Was this experiment performed by you or is this published data?
Despite the large volume of work, there are several comments
- - Keywords:
You can add a keyword: cotton (as a research object)
- Introduction
There is information about the OXS3 family of proteins and genes associated with them, but there is not enough information about cotton: how many species of cotton there are, why four species of cotton were chosen. Information about the cotton genome and species of cotton needs to be added.
- 2. Materials and Methods
2.3 Chromosome localization and gene duplication analysis of cotton OXS3 genes
151 “Each sample was analyzed in three technical 151 replicates and three biological replicates”
It would be good to describe in more detail the material for this experiment: How many plants per sample were taken for study? At what stages of plant development were samples fixed for RNA extraction? Under what conditions were the plants grown? Were these conditions controlled? Were samples fixed for each developmental stage at the same time for each plant?
- Results
276 “Aiming at screening the potential OXS3 genes that could regulate the specific biological processes, we analyzed the expression levels of OXS3 genes in G.hirsutum (GhOXS3-1 to GhOXS3-22) based on the published transcriptome data”
Please clarify, is this published data in Figure 6? In the case of published data, a link to the database must be provided
315 Five genes each of the OXS3 family members in Group 1 and Group 3 of G. hirsutum 315 were randomly selected for qRT-PCR analysis to examine the expression patterns of OXS3 316 genes under drought (PEG), salt (NaCl), cold (4°C), and heat (37°C) treatments at 0, 6, 12, 317 and 24 hours.
Describe in detail: how you treated the plants (PEG, sail, cold and heat)? 0 hours of treatment - is this a control?
336 –Figure 8.
Тhe ordinate axis labels are poorly visible
4. Discussion
433 “Overall, although there was no conclusive evidence linking OXS3 genes to early ovule development, these candidate genes might provide new insights into their potential functions in future studies”.
You have performed a large bioinformatics analysis of databases. Based on your data, can you suggest candidate genes that are associated with abiotic stress and/or any specific treatment (PEG, heat, cold, salt)?
Author Response
Comments 1: The Authors conducted comprehensively perform of the oxidative stress gene family in four Gossypium species. A large amount of data was summarized and bioinformatics analysis of evolutionary relationships, structural features, chromosomal localizations, collinearity, cis-acting elements was performed, and expression patterns were systematically studied using databases. These studies are relevant for further practical application in cotton breeding.
A big data was analyzed, but the contribution of the authors themselves is not always clear, for example, Figure 6: the data provided by the authors is a bioinformatics analysis of databases or data from a database is presented. Or in the case of qRT-PCR analysis of GhOXS3 expression under different stress conditions. Was this experiment performed by you or is this published data?
Response 1: Thank you very much for your worthy suggestion, and we are deeply sorry for not describing clearly on these results. In order to solve this problem, we have added one section in the Martials and Methods, namely 2.4 Analyses of tissue-specific, development-related, and respoding-adversity expression. This paragraph describes the details how we obtained the FPKM data of cotton OXS3 genes from the online database COTTONOMICS (http://cotton.zju.edu.cn/), performed the normalization processing by Z-score algorithm, and drew the heatmap to reflect the expression-pattern differences. Certainly, Figure 6 and Figur 7 were constructed by the published transcriptome data, and qRT-PCR experiments were utilized to verify the expression patterns responding to multiple abiotic stresses in Figure 8.
Comments 2: Despite the large volume of work, there are several comments - Keywords: You can add a keyword: cotton (as a research object)
Response 2: Thank you very much for your worthy suggestion. We have added the cotton as the keyword in the revised manuscript, which will be conducive to concentrating on the cotton species.
Comments 3: Introduction: There is information about the OXS3 family of proteins and genes associated with them, but there is not enough information about cotton: how many species of cotton there are, why four species of cotton were chosen. Information about the cotton genome and species of cotton needs to be added.
Response 3: Thank you very much for your worthy suggestion. We have added three paragraphs in the revised manuscript (Line 97-105), which could help us explain why we chose the four representative species in this study.
Comments 4: Materials and Methods 2.3 Chromosome localization and gene duplication analysis of cotton OXS3 genes, Line 151: “Each sample was analyzed in three technical 151 replicates and three biological replicates”.It would be good to describe in more detail the material for this experiment: How many plants per sample were taken for study? At what stages of plant development were samples fixed for RNA extraction? Under what conditions were the plants grown? Were these conditions controlled? Were samples fixed for each developmental stage at the same time for each plant?
Response 4: Thank you very much for your worthy suggestion. We have revised the relative section in the modified manuscript, and added the descriptions on the material planting, cultivating conditions, stress treatments, and sample collection in Line 168-183.
Comments 5: Results Line 276 “Aiming at screening the potential OXS3 genes that could regulate the specific biological processes, we analyzed the expression levels of OXS3 genes in G.hirsutum (GhOXS3-1 to GhOXS3-22) based on the published transcriptome data”. Please clarify, is this published data in Figure 6? In the case of published data, a link to the database must be provided.
Response 5: Thank you very much for your careful and valuable reviewing. We have added one paragraph entitled “2.4 Analyses of tissue-specific, development-related, and respoding-adversity expression”, which described the details how we utilized the online database COTTONOMICS (http://cotton.zju.edu.cn/), performed the normalization processing by Z-score algorithm, and drew the heatmap to reflect the expression-pattern differences. Certainly, Figure 6 and Figur 7 were constructed by the published transcriptome data, and qRT-PCR experiments were utilized to verify the expression patterns responding to multiple abiotic stresses in Figure 8.
Comments 6: Line 336 –Figure 8. The ordinate axis labels are poorly visible
Response 6: Thank you very much for your worthy suggestion, and we have revised the Figure 8 in the modified manuscript.
Comments 7: Discussion, Line 433 “Overall, although there was no conclusive evidence linking OXS3 genes to early ovule development, these candidate genes might provide new insights into their potential functions in future studies”. You have performed a large bioinformatics analysis of databases. Based on your data, can you suggest candidate genes that are associated with abiotic stress and/or any specific treatment (PEG, heat, cold, salt)?
Response 7: hank you very much for your careful and valuable reviewing, and we have added the relative sentences (Line 499-507) in the revised manuscript. After comprehensive analyses on the cotton OXS3 genes under the developing ovules and fibers, and in response to multiple abiotic stresses, GhOXS3-3 gene could be chosen to perform the overexpressing or knock-out experiments in Arabidopsis and cotton to validate its function combined with the phenotypes of the transgenic plants under various abiotic stress conditions. We believe that the over-expressed or silenced materials will ultimately lead to the development of new cotton varieties with enhanced stress resistance.
Reviewer 3 Report
Comments and Suggestions for Authors
General comment
This manuscript provides a comprehensive genome-wide analysis of the OXS3 gene family in four representative species of cotton (Gossypium hirsutum, Gossypium barbadense, Gossypium arboreum, and Gossypium raimondii). The authors explore the evolutionary relationships, gene structure, chromosomal locations, and expression patterns of OXS3 genes, particularly in response to abiotic stresses such as cold, heat, salt, and drought. Furthermore, the study employs both bioinformatics and quantitative real-time PCR (qRT-PCR) approaches to assess the potential role of OXS3 genes in cotton development and stress tolerance. This research adds valuable insight into cotton molecular genetics, particularly for breeding stress-tolerant varieties.
The novelty of this study lies in its comprehensive analysis of the OXS3 gene family across multiple cotton species, a family that has not been extensively studied in cotton before. The genome-wide identification, coupled with detailed expression analysis in response to abiotic stresses, provides new insights into how these genes might contribute to cotton stress tolerance and fiber development. This work is particularly novel as it links the OXS3 family to traits important for cotton breeding, offering potential applications for improving crop resilience.
The methodology employed in the study is solid, relying on well-established bioinformatics tools for genome-wide identification and structural analysis, as well as qRT-PCR to validate gene expression under stress conditions. The use of high-throughput RNA sequencing data further strengthens the reliability of the findings. However, additional experimental validation (e.g., functional studies using knock-out or overexpression lines) would be necessary to fully establish the roles of specific OXS3 genes in stress tolerance.
The study effectively demonstrates the relevance of OXS3 genes in responding to abiotic stresses and highlights candidate genes that could be targeted in cotton breeding programs. The identification of stress-responsive elements in promoter regions adds depth to the analysis. However, the manuscript could be more effective in translating these findings into practical applications by further discussing how these genes can be utilized in breeding programs or genetic engineering.
In summary, this manuscript presents a detailed and novel exploration of the OXS3 gene family in cotton. The study is scientifically sound and offers significant contributions to the field of cotton genetics and stress tolerance. While the findings are reliable and potentially impactful, further functional studies would enhance the overall strength and applicability of the research. This paper provides the readers of Biology with a valuable insight into tolerance to the environmental stress. However, the authors need to revise the manuscript as follows.
Specific Comments and Suggestions for Revisions
Introduction:
Comment: The introduction provides a good overview of OXS3 gene function but lacks clarity in linking these genes to cotton breeding and the potential applications. Some sentences seem overly long and complex, making it harder to follow the scientific logic.
Suggestion: Clarify the practical importance of studying the OXS3 family in cotton by emphasizing how your findings could directly aid in breeding stress-tolerant cotton varieties. Simplify the sentence structure in some areas for better readability.
Materials and Methods:
Comment: This section is comprehensive but contains redundant details that can be condensed. For instance, repeating website URLs and software tools multiple times detracts from the flow.
Suggestion: Remove or consolidate redundant information, such as multiple mentions of software and database sources. Focus on describing the unique methodologies applied in this study.
Results:
Genome-wide Identification
Comment: This section is well-structured but could be enhanced by providing a clearer explanation of why the identified OXS3 proteins are significant. The phylogenetic tree and the accompanying description are good but could benefit from a more thorough comparison with related studies.
Suggestion: Provide more interpretation of the phylogenetic tree results, particularly in terms of functional divergence between groups. Highlight specific genes of interest that may have novel functions in cotton.
Expression Patterns
Comment: The explanation of expression patterns is detailed, but the connection to real-world applications (e.g., cotton breeding for stress resistance) is not emphasized enough.
Suggestion: Explicitly state how the differential expression of OXS3 genes under stress conditions can be used to improve cotton resilience. More graphical summaries (e.g., visual representation of expression levels) could help non-specialist readers.
Discussion:
Comment: The discussion is somewhat lengthy and could benefit from more focus. It covers various aspects but does not deeply engage with the potential for translating these findings into practical cotton breeding applications.
Suggestion: Shorten the discussion by focusing on the most important and novel aspects of your findings. Bring out the practical implications of this research more clearly—how does it contribute to the next steps in cotton genetic research?
Figures and Tables:
Comment: Figures and tables are generally clear but could benefit from more descriptive captions. Some graphs, especially the heatmaps, lack context regarding the data's significance.
Suggestion: Add more detail to figure captions explaining why each result matters. For the heatmaps, briefly describe the key findings directly in the caption to guide the reader.
Author Response
Comments 1: In Introduction, Comment: The introduction provides a good overview of OXS3 gene function but lacks clarity in linking these genes to cotton breeding and the potential applications. Some sentences seem overly long and complex, making it harder to follow the scientific logic.
Suggestion: Clarify the practical importance of studying the OXS3 family in cotton by emphasizing how your findings could directly aid in breeding stress-tolerant cotton varieties. Simplify the sentence structure in some areas for better readability.
Response 1: Thank you very much for your careful and valuable reviewing, and we have revised the complex sentences all through the manuscript. Also, plenty of the repeated sentences were deleted in the revised manuscript, and try our best to improve the legibility.
Comments 2: In Materials and Methods, Comment: This section is comprehensive but contains redundant details that can be condensed. For instance, repeating website URLs and software tools multiple times detracts from the flow.
Suggestion: Remove or consolidate redundant information, such as multiple mentions of software and database sources. Focus on describing the unique methodologies applied in this study.
Response 2: Thank you very much for your careful and valuable reviewing. We have deleted the repeating websites and software tools in the Results, and keep those in the Materials and Methods. After concentrating on the unique methodologies applied in this study, the legibility of our article has been greatly improved.
Comments 3: In Results, Comment: This section is well-structured but could be enhanced by providing a clearer explanation of why the identified OXS3 proteins are significant. The phylogenetic tree and the accompanying description are good but could benefit from a more thorough comparison with related studies.
Suggestion: Provide more interpretation of the phylogenetic tree results, particularly in terms of functional divergence between groups. Highlight specific genes of interest that may have novel functions in cotton.
Response 3: Thank you very much for your careful and valuable reviewing. We have added the relative sentences in the phylogenetic analysis of Results (Line 207-210), which will be conducive to explaining why more cotton OXS3 members could provide more functional divergences.
Comments 4: As for Expression patterns, Comment: The explanation of expression patterns is detailed, but the connection to real-world applications (e.g., cotton breeding for stress resistance) is not emphasized enough.
Suggestion: Explicitly state how the differential expression of OXS3 genes under stress conditions can be used to improve cotton resilience. More graphical summaries (e.g., visual representation of expression levels) could help non-specialist readers.
Response 4: Thank you very much for your worthy suggestion. We have added one paragraph entitled “2.4 Analyses of tissue-specific, development-related, and respoding-adversity expression”, which described the details how we utilized the online database COTTONOMICS (http://cotton.zju.edu.cn/), performed the normalization processing by Z-score algorithm, and drew the heatmap to reflect the expression-pattern differences. Certainly, the published RNA-seq data were only utilized to predict the potential functions during development and growth, and in response to multiple stresses. In order to verify the various expression patterns responding to stresses, qRT-PCR experiments were conducted in this study to screen the candidate OXS3 genes.
Comments 5: In Discussion, Comment: The discussion is somewhat lengthy and could benefit from more focus. It covers various aspects but does not deeply engage with the potential for translating these findings into practical cotton breeding applications.
Suggestion: Shorten the discussion by focusing on the most important and novel aspects of your findings. Bring out the practical implications of this research more clearly—how does it contribute to the next steps in cotton genetic research?
Response 5: Thank you very much for your careful and valuable reviewing. We have added the relative sentences (Line 499-507) in the revised manuscript. After comprehensive analyses on the cotton OXS3 genes under the developing ovules and fibers, and in response to multiple abiotic stresses, GhOXS3-3 gene could be chosen to perform the overexpressing or knock-out experiments in Arabidopsis and cotton to validate its function combined with the phenotypes of the transgenic plants under various abiotic stress conditions. We believe that the over-expressed or silenced materials will ultimately lead to the development of new cotton varieties with enhanced stress resistance.
Comments 6: As for Figures and Tables, Comment: Figures and tables are generally clear but could benefit from more descriptive captions. Some graphs, especially the heatmaps, lack context regarding the data's significance.
Suggestion: Add more detail to figure captions explaining why each result matters. For the heatmaps, briefly describe the key findings directly in the caption to guide the reader.
Response 6: Thank you very much for your worthy suggestion. We have replaced Figure 7 and Figure 8, since their arrangements were not reasonable. Meanwhile, we also added the notes in the heatmaps, which will be conducive to readers for comprehending the results.